# Optimizing Transformers with Approximate Computing for Faster, Smaller and more Accurate NLP Models

## Abstract

Transformer models have garnered a lot of interest in recent years by delivering state-of-the-art performance in a range of Natural Language Processing (NLP) tasks. However, these models can have over a hundred billion parameters, presenting very high computational and memory requirements. We address this challenge through Approximate Computing, specifically targeting the use of Transformers in NLP tasks. Transformers are typically pre-trained and subsequently specialized for specific tasks through transfer learning. Based on the observation that pre-trained Transformers are often over-parameterized for several downstream NLP tasks, we propose a framework to create smaller, faster and in some cases more accurate models. The key cornerstones of the framework are a Significance Analysis (SA) method that identifies components in a pre-trained Transformer that are less significant for a given task, and techniques to approximate the less significant components. Our approximations include pruning of blocks, attention heads and weight groups, quantization of less significant weights and a low-complexity sign-matching based attention mechanism. Our framework can be adapted to produce models that are faster, smaller and/or more accurate, depending on the user's constraints. We apply our framework to seven Transformer models, including optimized models like DistilBERT and Q8BERT, and three downstream tasks. We demonstrate that our framework produces models that are up to $4\times$ faster and up to $14\times$ smaller (with less than 0.5% relative accuracy degradation), or up to 5.5% more accurate with simultaneous improvements of up to $9.83\times$ in model size or $2.94\times$ in speed.

## 1 Introduction

Transformer networks with hundreds of billions of parameters, such as T5 (Raffel et al. (2019)), Megatron (Shoeybi et al. (2019)), BERT (Devlin et al. (2019)), GPT-2 (Radford et al. (2019)) and GPT-3 (Brown et al. (2020)), have achieved state-of-the-art performance in several Natural Language Processing tasks. Model sizes are expected to grow further in the future as increasing the number of parameters has been shown to improve performance. For instance, increasing the number of parameters from 1.5B to 175B enabled a reduction in perplexity for Language Modelling (Penn Treebank) from 35.8 in GPT-2 to 20.5 in GPT-3. This makes it computationally challenging to train Transformers as well as perform inference using them. The challenges associated with training these models are alleviated through the (re-)use of pre-trained models that are subsequently fine-tuned for different tasks. Consequently, these models incur a major one-time cost in computational resources, time and energy during the pre-training process, but the repeated fine-tuning for individual downstream tasks is performed at a considerably lower cost.

However, performing inference using fine-tuned Transformer models continues to remain a challenge because of the large amount of storage and compute operations required. Prior research efforts have explored different techniques for improving the efficiency of Transformer inference. However, several of the proposed approaches either require training the network completely from scratch (which is extremely compute and memory-intensive), or cause significant degradation in accuracy on the downstream task. In this work, we overcome these limitations by exploiting the transfer learning step in Transformers to produce individually optimized models for the different

downstream tasks, using techniques that do not require training from scratch and maintain or improve accuracy levels.

From the runtime and memory breakdown of Transformers (Fig. 1), we observe that the most time-consuming and memory-intensive operations in a Transformer are the self-attention (ATTN) blocks, which are used to identify and form relationships between the different tokens in text, and the feed-forward neural network blocks (FFN blocks) in the Transformer layers. These blocks together account for more than 85% of the inference time (and more than 75% of the model's parameters). We accordingly optimize the execution of these two components in our approach. The self-attention component dominates the execution time and memory size for longer context lengths as its operation scales quadratically in time and memory with sequence length. Some previous works (Kitaev et al. (2020), Ye et al. (2019)) have addressed this issue, accelerating training and inference of Transformers when large context lengths are used. However, they suffer from significant overheads and slowdowns in applications with smaller context lengths, such as question answering, where questions and answers are usually short, in the order of a few hundred tokens. Our approach, on the other hand, performs well across context lengths, size of hidden layers, number of layers and other network characteristics.

The pre-training of Transformer models with some initial objective (most commonly predicting masked words in a large text corpus) and the subsequent fine-tuning on a downstream task leads to highly over-parameterized models for many downstream tasks (Michel et al. (2019)), providing ample opportunities for approximations. As these models grow larger, such opportunities are expected to increase even further. We observe that for a given downstream task, some parts of the pre-trained Transformer are more significant to obtain good accuracy, while other parts are less important or unimportant. In order to exploit this observation in a principled manner, we introduce a framework to introduce approximations while fine-tuning a pre-trained Transformer network, optimizing for either size, latency, or accuracy of the final network. We perform and apply significance analysis in a hierarchical manner, first pruning entire blocks, followed by attention heads, and finally pruning weight groups. We achieve further gains by also allowing elements that cannot be pruned to be approximated by other techniques. We specifically apply two forms of approximations, depending on the element type. For weights, we utilize quantization. For the self-attention operation, we replace the scaled dot product attention mechanism with a novel sign matching-based attention mechanism.

We summarize our main contributions as follows:

- We introduce a framework for creating fine-tuned models from pre-trained Transformer models that are optimized for various metrics (size, latency, accuracy).

- We incorporate multiple heuristics in the framework, such as hierarchical processing, model-driven insights, and run-time based ordering of elements.

- We propose a significance analysis technique to identify the importance of each element of the pre-trained Transformer for a given downstream task. We use this technique to prune entire blocks, attention heads, and weight groups and to guide the quantization of low-importance weights.

- We propose a low-complexity attention mechanism, sign matching, in order to approximate dot product attention in the less significant attention layers.

- Across a suite of different Transformer networks, including previously proposed optimized networks, we demonstrate that our techniques produce models that are up to $4\times$ faster and up to $14\times$ smaller (with less than 0.5% relative accuracy degradation), or up to 5.5% more accurate with simultaneous size and latency improvements.

## 2 RELATED WORK

Given the effectiveness and popularity of Transformer models, several techniques have been proposed to overcome their computational and memory challenges, and to accelerate inference using these models. Most of these works directly pre-train efficient models from scratch. For example, DistilBERT (Sanh et al. (2019)), MobileBERT (Sun et al. (2020)) and TinyBERT (Jiao et al. (2019)) use knowledge distillation to train smaller and faster networks using the original network as a teacher. LayerDrop (Fan et al. (2020)) randomly drops layers during pre-training, thereby enabling

their dropping during inference. SchuBERT (Khetan & Karnin (2020)) learns the optimal sizes of the BERT elements during pre-training. Lite Transformer (Wu et al. (2020)) uses Long-Short Range Attention to speed up the self-attention operation, with different attention heads attending to local and global context. Depth-adaptive Transformer (Elbayad et al. (2020)) and DeeBERT (Xin et al. (2020)) modulate Transformer depth depending on the complexity of each input sample using gating functions that are trained along with the model. AlBERT (Lan et al. (2020)) uses factorized embeddings and cross-layer parameter sharing. These works are orthogonal to ours, as the models that they produce are still subsequently fine-tuned for downstream tasks. We demonstrate using DistilBERT, AlBERT and LayerDrop as examples that these optimized networks still have significant opportunities that our techniques can take advantage of.

Other works (including ours) aim to improve the inference efficiency of Transformers using techniques that do not require training new models from scratch. Among these, PoWER-BERT (Goyal et al. (2020)), which eliminates redundant word vectors from the model without removing any parameters, and Q8BERT (Zafrir et al. (2019)), which quantizes all weights and activations in the model to 8-bit integers through the use of Quantization-Aware Training at fine-tuning time, are orthogonal and complementary to our work. Poor Man's BERT (Sajjad et al. (2020)) evaluates several layer-dropping techniques that do not require re-training. Compared to layer-dropping techniques that do not require re-training, our techniques produce models that are up to 20% more accurate at comparable inference speed, and this is especially true when working with highly optimized baselines such as Q8BERT. Our framework can also be adapted to satisfy a wide range of user constraints.

## 3 PRELIMINARIES

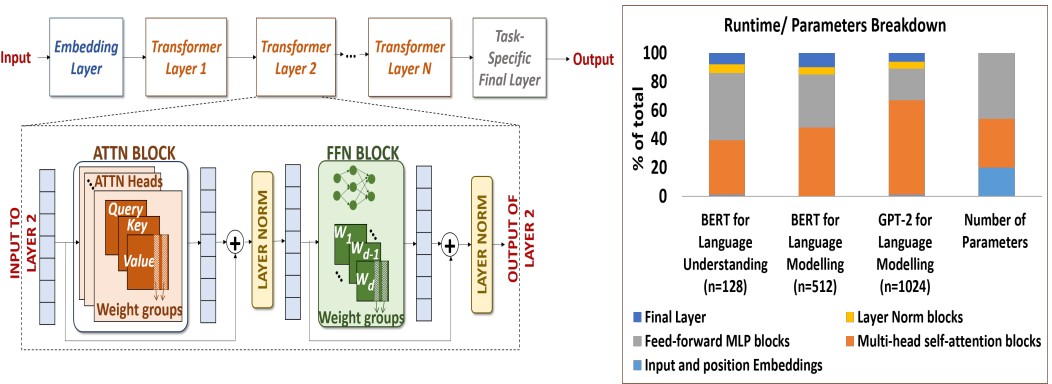

Figure 1: **[Left] A typical Transformer architecture and its elements. [Right] Runtime/parameter-count breakdown of Transformers.** At smaller context lengths, the feed-forward neural network is the time-dominant operation. As context length increases, self-attention becomes the time-dominant operation. In both cases, ATTN and FFN blocks together account for >75% of total parameters, and >85% of the total runtime on an NVIDIA GTX 1080 Ti GPU.

A Transformer (Fig. 1) consists of an embedding layer, followed by multiple transformer layers stacked together, and a task-specific final layer. A transformer layer consists of the multi-headed self-attention operation (ATTN block), followed by a feed-forward neural network (FFN block) with layer norm operations at the input and output of the layer. In this work, we define the *elements* of a Transformer to include different levels of granularity, *i.e.*, ATTN blocks, FFN blocks, Attention Heads and Weight Groups. We define Weight Groups only along dimensions that do not impact the shape of the output of the block when these groups are removed.

The self-attention operation takes as input a sequence $n$ of vectors X, and computes three matrices, Query = $X \times W_q$, Key = $X \times W_k$ and Value = $X \times W_v$. Then, the output of the self-attention operation is computed as $Y = softmax((Query \times Key^T) + attention\_mask) \times Value$. For auto-regressive models, tokens are not allowed to attend to future tokens. Hence, an attention mask is applied before the softmax operation, setting attention scores with future tokens to a very large negative number, which becomes zero after the softmax operation. This operation has multiple

"attention heads" working in parallel on the input sequence, where each head has its own set of parameters to compute the query, key and value matrices. The independent attention outputs are concatenated and transformed into the expected output dimensions. The self-attention operation scales quadratically in time and memory with sequence length $n$ since $Query \times Key^T$ has $n^2$ entries.

# 4 DESIGN METHODOLOGY

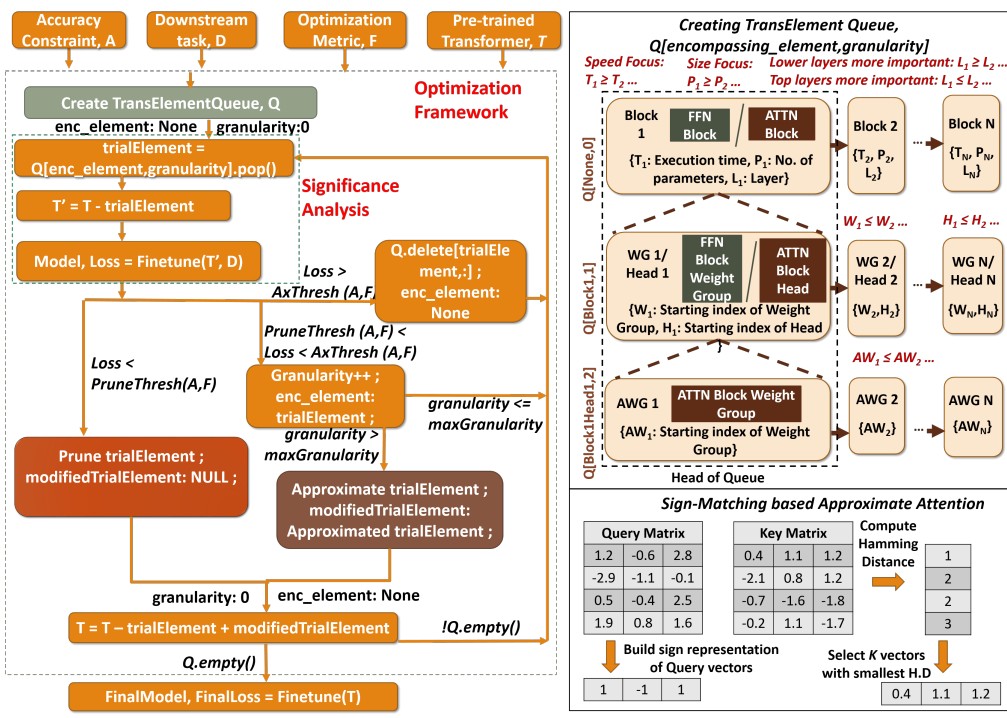

Figure 2: **Illustration of the Transformer Approximation Methodology**. Sign Matching is illustrated with **K**=1

We propose a framework for producing fine-tuned Transformer models that are optimized for a specific metric (speed, model size, or accuracy). Fig. 2 presents an overview of the proposed framework. As shown in the figure, the inputs to the framework are a pre-trained Transformer model, the fine-tuning dataset, the goal of optimization (speed, size or accuracy) and acceptable accuracy loss (when optimizing for speed or size). The framework has three major components: (i) a set of heuristics used to build an ordered queue of elements (TransElements) to be considered, (ii) a significance analysis method to identify insignificant elements in a pre-trained Transformer and (iii) a set of techniques to prune or approximate the insignificant elements. The framework proceeds in an iterative manner. That is, we first start with the original Transformer. We then remove an element from the TransElements queue, analyze its significance, and apply pruning/approximation techniques to the element. This results in new Transformer, where the element is replaced by the pruned or approximated version. This modified Transformer is then used as the baseline for the next iteration. After processing all of the identified elements, we fine-tune on the downstream task for the same number of epochs as the baseline model to obtain the final, optimized model. A detailed description of our methodology for approximating Transformers is presented in Fig. 2 and in Algorithm 4. In the following subsections, we further describe our techniques for generating the ordered queue TransElements, followed by the significance analysis method, and finally the pruning and approximation techniques for different Transformer elements.

**TransElement Ordered Queue.** In order to optimize a given model, we would ideally want to characterize the significance of each and every parameter in the model, rank them in order of importance, and finally prune/approximate only the least significant parameters, as in Molchanov et al. (2017).

However, Transformers have billions of parameters, making this process computationally infeasible. In addition, previously proposed techniques that can efficiently estimate the importance of each parameter, such as using Taylor expansion, are not applicable. This is because the {approximate, fine-tune, approximate} cycle does not work for Transformers during fine-tuning, since they very quickly overfit the training data for the downstream task (usually within 5 epochs). We take advantage of hierarchical structure of Transformers and consider them in a hierarchical manner, ordered by increasing granularity. Specifically, we place entire FFN and ATTN blocks earlier in the queue, followed by heads, and finally weight groups. Through this ordering, we are able to quickly eliminate large numbers of parameters from further consideration, speeding up future iterations of the framework. For example, eliminating a single FFN block in the BERT-Base model removes 5.6% of all parameters under consideration. To further reduce the number of elements under consideration, we also dynamically remove elements from the queue if they are encompassed by a high-importance block. For example, if a given ATTN block is determined to be of high importance, we remove all heads and weight groups within that block from the TransElement queue.

Since the framework iterates through the entries of the TransElement queue sequentially, its efficacy is dependent on the ordering of the elements at each level of granularity. In order to minimize the run-time of the framework, we provide two additional heuristics to guide the ordering of elements. First, we use the unique linguistic properties captured by the different Transformer layers (Jawahar et al. (2019)). These properties depend on both the Transformer and the downstream task under consideration, since different tasks require different types of linguistic knowledge. For example, top layers usually have low significance for Language Understanding tasks, since long-range dependency information is not required for most tasks (for example, sentiment analysis requires only local context). Hence, we place the final layer at the front of the queue, and work our way backwards towards the first layer, since blocks in the final layers are more likely to be removed, thereby speeding up future iterations. Second, we use a run-time (or parameter-count) aware ordering of the TransElements, such the most time consuming blocks (or blocks with the most parameters) are likely to be removed earlier in the algorithm. For example, at large context lengths, we start with the ATTN blocks in all layers before moving on to the FFN blocks, and vice-versa at small context lengths. This has the dual benefit of producing highly optimized models for inference, as well as speeding up the significance analysis process by eliminating time-consuming blocks early and making further iterations faster. Algorithm 1 and Fig. 2 describe the process of creating the TransElement Queue. The utility of this framework and the heuristics used are discussed in Appendix C.

**Significance Analysis.** To determine the significance of each Transformer element, we first fine-tune the original Transformer model for the given downstream task to obtain the baseline loss. We then use this baseline loss, along with the provided acceptable accuracy degradation, to generate a set of loss thresholds that determine whether a given element is of low importance and therefore can be pruned/approximated. This is a one-time step and performed globally for all elements in the TransElements queue. Then, for the element under consideration in each iteration of the framework, we compute the loss of the current Transformer model with the element removed. We then compare this loss to the thresholds determined above. The exact thresholds used are dependent on the optimization metric: speed, size, or accuracy. If we are optimizing the network for speed or size, we prune the element under consideration if the training/validation loss upon removing it from the Transformer is less than the pruning threshold. If we are optimizing for accuracy, we prune the element only if the training/validation loss when it is removed is less than the minimum loss seen thus far during the optimization process, since the goal is to find a model with minimum loss. Similarly, we apply approximations if the loss with the element removed from the Transformer is greater than the pruning threshold but lower than the approximation threshold. Algorithm 2 describes Significance Analysis.

**Pruning and Approximating.** As evident from Section 3, the structure and functionality of ATTN blocks differ significantly from that of FFN blocks in a Transformer. We accordingly adopt different strategies for approximating them, as described below. But pruning an entire ATTN or FFN block is effectively the same as it simply involves using the skip connection to bypass that block. The pruning strategies for the FFN and ATTN blocks are illustrated in Fig. 4 and Fig. 5.

**Pruning Weight Groups within approximable FFN Blocks.** Consider an approximable FFN block that performs the transformation $\mathbb{R}^{n \times d} \times \mathbb{R}^{d \times y} \to \mathbb{R}^{n \times y}$, with weight groups defined along the $d$ dimension (($d/W$) weight groups of ($W$) weights each, where $W$ is a hyperparameter that defines the granularity of approximations). When optimizing models for accuracy, we remove weight groups

only if doing so results in a reduction in the model loss. When optimizing for size, we remove weight groups that maintain loss within the pruning threshold when removed. When optimizing for speed, however, removing weight groups with low significance from arbitrary locations does not help, since it introduces unstructured sparsity in the weight matrix that can be difficult to exploit to achieve speedups. Instead, we impose structure on our pruning. Specifically, we use a "greedy shrinking" algorithm that finds the largest number of weight groups that can be removed while maintaining loss below the threshold, such that the weight groups that remain in the model form a contiguous block. We first start from the bottom (weight group 0), work our way up and remove as many weight groups as possible while staying within the loss threshold. We then start from the top (weight group $d/W$), work our way down and remove as many weight groups as possible while staying within the loss threshold. When this process is completed, the weight groups that remain form a contiguous dense block, enabling speedups on all hardware platforms. Since weight groups are removed along the "hidden" dimension $d$, our methods do not change the shape of the output of this block; instead, we are simply "shrinking" the effective hidden dimension size through structured pruning.

**Quantizing Weight Groups within approximable FFN and ATTN Blocks.** When optimizing the Transformer for size, we also quantize weight values within weight groups for which the loss lies between the pruning and approximation thresholds. We use uniform quantization with Quantization-Aware Training proposed in Q8BERT (Zafrir et al. (2019)) within our hierarchical framework to quantize insignificant weight groups to lower precisions. This reduces the memory requirements of those weight groups but does not improve the execution time as the computations are still performed at the baseline precision.

**Pruning ATTN heads and Weight Groups within approximable ATTN Blocks.** We divide the multi-headed self-attention operation into two main steps. In the first step, we compute the Query, Key and Value matrices by multiplying the input to this layer with the corresponding weight matrices ($\mathbb{R}^{n \times d} \times \mathbb{R}^{d \times y} \to \mathbb{R}^{n \times y}$), and then reshape them into multiple attention heads ($\mathbb{R}^{n \times y} \to \mathbb{R}^{n \times h \times (y/h)}$). Our approach to pruning this step is exactly the same as for the FFN blocks, where we iteratively prune weight groups along the $d$ dimension using our shrinking algorithm. In the second step, we compute the "attention output" as $Y = softmax((Query \times Key^T) + attention\_mask) \times Value$. To optimize this step, we apply two techniques. Firstly, we identify insignificant attention heads, and prune them from the model. However, removing attention heads changes the shape of the output of this layer. We overcome this by keeping track of the pruned heads, and padding the output with zeros in the corresponding locations. In spite of this overhead, we still manage to achieve significant speedup from this approximation technique since pruning heads makes multiple downstream operations (computing the attention scores, applying softmax to the attention scores, and computing the final score) considerably faster. Therefore, we do not use our greedy shrinking method, but rather rely on unstructured pruning as it allows for greater pruning which further benefits the downstream operations. Secondly, we dynamically reduce the size of the key and value matrices by pruning weight groups from the same location along the $n$ dimension in both matrices, based on sign matches with the query vectors. This again makes multiple downstream operations considerably faster and does not change the shape of the output of the pruned block.

**Approximating self-attention within approximable ATTN Blocks.** We observe that the "attention scores" matrix is highly sparse, especially after the softmax operation. This sparsity implies that most of the dot products between the query and the key are unnecessary. Thus, we would ideally like to perform the attention operations for the query vectors that give highest dot-product with each key vector efficiently without explicitly performing all of the dot products. To this end, we propose replacing the $O(n^2)$ dot product-based attention mechanism with a linear-time sign-matching-based mechanism in approximable ATTN blocks. Sign-matching attention (SM) is based on the idea that key vectors whose signs match with the largest number of query vectors will have high dot-products with maximum number of query vectors. However, it is expensive to compute a sign match for all pairs of query-key vectors, as this will grow quadratically. Instead, we employ a low-cost approximation. For each column of the query matrix, we identify if more number of vectors will have a positive or negative number in that column. This becomes the representative sign in that column for all the query vectors. Each key vector is then scored by how well the sign of each of its elements matches with the sign of the representative query vector by computing the Hamming distance between the two sign vectors. This score is used to select the top $K$ key vectors. As a result, we reduce the number of computations required to score each key vector (and the overall complexity) from $O(n^2)$ to $O(n)$. Sign matching is illustrated in Fig. 2, and explained in detail in Appendix B. As this

approximation does not increase the accuracy of the models nor decrease the number of parameters, they are only applied when optimizing the fine-tuned models for speed.

## 5 EXPERIMENTS AND RESULTS

We implement our techniques within Huggingface's Transformers library in PyTorch (Wolf et al. (2019)). We use Intel AI's NLP Architect for experiments on Q8BERT. The experiments were performed on a GeForce RTX 2080 Ti GPU with 11GB memory. All results reported are the average of 10 runs with random seeds after 3 epochs of fine-tuning on the dev set, unless otherwise specified. When reporting results on the development set, for the larger datasets (like MNLI), we create a validation set using a small subset ( 15%) of the training data. We use the loss on this set to characterize the significance. On the smaller datasets (like WNLI), there isn't enough data to get a meaningful training-validation split. Hence, we directly use the loss on the training set. When reporting results on the test set, we use loss on the development set to characterize significance. Detailed descriptions of the tasks and Transformers used in our experiments is given in Appendix E. Additional results on the GLUE test set are presented in Appendix F.

**Primary Results.** We present results on GLUE (Wang et al. (2019)) in Table 1, SQUADv1.1 (Rajpurkar et al. (2016)), and Penn Treebank (Marcus et al. (1994)) in Table 2. When optimizing for speed, we aim to reduce inference time as much as possible while maintaining at least 99.5% of baseline accuracy. While the focus in these experiments is on speed, we find that our framework still leads to models that are `1.29×-10.65×` smaller due to TransElements being dropped from the pre-trained model. On the other hand, when optimizing for size, we focus on reducing the model size as much as possible while maintaining the <0.5% accuracy degradation constraint. We use uniform quantization with Quantization-Aware Training proposed in Q8BERT

| | CoLA | MNLI | MRPC | QNLI | QQP | RTE | SST-2 | STS-B | WNLI | Average |
|---|---|---|---|---|---|---|---|---|---|---|
| **XLNet** | | | | | | | | | | |
| Baseline | 55.2 | 86.76 | 87.05 | 90.12 | 90.78 | 54.42 | 93.01 | 87.69 | 56.34 | 77.93 |
| Accuracy Focus | 57.89 | 86.97 | 87.61 | 90.18 | 91.11 | 59.88 | 93.62 | 87.93 | 56.48 | 79.08 |
| Speed Focus (Speedup) | 54.91 (3.81X) | 86.54 (1.35X) | 86.87 (1.63X) | 89.79 (1.35X) | 90.65 (2.01X) | 54.15 (3.11X) | 93.38 (1.68X) | 87.52 (1.9X) | 56.33 (4.22X) | 77.79 (2.34X) |
| Size Focus (Compression) | 54.93 (14.06X) | 86.49 (5.89X) | 86.88 (6.04X) | 89.91 (5.98X) | 90.68 (9.5X) | 54.32 (9.96X) | 93.46 (9.1X) | 87.51 (4.88X) | 56.32 (14.08X) | 77.83 (8.85X) |
| **Q8BERT** | | | | | | | | | | |
| Baseline | 52.85 | 83.73 | 85.05 | 91.19 | 90.65 | 64.26 | 92.32 | 88.09 | 56.11 | 78.25 |
| Accuracy Focus | 55.45 | 83.92 | 85.89 | 91.26 | 91.19 | 64.8 | 92.76 | 88.97 | 56.72 | 78.99 |
| Speed Focus (Speedup) | 53.91 (1.45X) | 83.42 (1.43X) | 85.02 (1.33X) | 90.79 (1.4X) | 90.36 (1.66X) | 63.91 (1.41X) | 92.48 (1.43X) | 87.88 (1.51X) | 56.34 (2.85X) | 78.23 (1.61X) |
| Size Focus (Compression) | 53.57 (13.84X) | 83.56 (5.64X) | 85.03 (5.56X) | 90.91 (5.52X) | 90.48 (8.44X) | 63.97 (10.48X) | 92.42 (8.12X) | 87.94 (5.52X) | 56.28 (14.08X) | 78.24 (8.6X) |
| **DistilBERT** | | | | | | | | | | |
| Baseline | 50.25 | 82.04 | 84.07 | 88.72 | 89.92 | 61.01 | 90.48 | 86.49 | 56.33 | 76.59 |
| Accuracy Focus | 52.12 | 82.28 | 84.64 | 88.91 | 90.28 | 62.14 | 90.6 | 86.94 | 56.82 | 77.19 |
| Speed Focus (Speedup) | 50.06 (1.46X) | 81.84 (1.34X) | 84.06 (1.34X) | 88.83 (1.34X) | 89.86 (1.46X) | 61.73 (1.44X) | 90.25 (1.29X) | 86.34 (1.43X) | 56.33 (1.54X) | 76.57 (1.41X) |
| Size Focus (Compression) | 50.14 (7.08x) | 81.82 (4.62x) | 84.03 (4.66x) | 88.79 (4.59x) | 89.88 (6.12x) | 61.58 (6.02x) | 90.29 (4.21x) | 86.31 (6.36x) | 56.33 (7.11x) | 76.57 (5.6x) |
| **BERT** | | | | | | | | | | |
| Baseline | 53.21 | 84.43 | 86.11 | 90.68 | 91.06 | 64.25 | 93.23 | 88.13 | 56.11 | 78.58 |
| Accuracy Focus | 55.45 | 84.69 | 88.89 | 90.83 | 91.77 | 64.8 | 94.06 | 89.01 | 56.82 | 79.59 |
| Speed Focus (Speedup) | 53.91 (2.92x) | 84.28 (1.29x) | 86.01 (2.21x) | 90.41 (1.28x) | 90.87 (1.84x) | 64.32 (3.29x) | 93.04 (1.33x) | 87.99 (1.89x) | 56.33 (3.51x) | 78.57 (2.17x) |
| Size Focus (Compression) | 53.91 (13.16x) | 84.26 (5.19x) | 86.05 (6.61x) | 90.32 (5.22x) | 90.81 (8.36x) | 64.32 (10.88x) | 92.99 (8.01x) | 88.02 (5.08x) | 56.33 (14.08x) | 78.56 (8.5x) |

Table 1: **Results on GLUE.** We report Matthews correlation for CoLA, Pearson Correlation for STS-B and accuracy for all other tasks. We report only "matched" accuracy for MNLI.

(Zafrir et al. (2019)) within our hierarchical framework to quantize insignificant blocks, heads and weight groups to lower precisions. This leads to models that are smaller than and at least as fast as a uniform 8-bit integer quantized model such as Q8BERT (Table 1). Our results are competitive with QBERT (Shen et al. (2020)), while maintaining the advantages of uniform 8-bit quantization over the group-wise quantization proposed in QBERT. The compression is lowest for AlBERT since its parameters are shared across layers, and most of the compression is from quantization. While the focus in these experiments is on size, we find that our framework still leads to models that are `1.07×-3.26×` faster due to elements being dropped from the pre-trained model, with potential for much greater speedups on optimized 8-bit integer kernels. When optimizing for accuracy, the goal is to maximize the accuracy of the pre-trained Transformer model for any given downstream task. While the focus in these experiments is on accuracy, we find that our framework still leads to models that are `1.28×-9.83×` smaller and `1.03×-2.94×` faster due to TransElements being dropped from the pre-trained model.

Table 2: **[Left] Results on SQUAD v1.1.** We report the Exact Match score. The compression is lowest for AlBERT since parameters are shared across layers, and most of the compression is from quantization. **[Right] Results on Penn Treebank.** We report perplexity (lower is better).

| Network | Baseline | Accuracy Focus | Speed Focus (Speedup) | Size Focus (Compression) |
|---|---|---|---|---|
| AlBERT | 80.97 | 81.32 | 80.66 (1.42X) | 80.71 (1.28X) |
| XLNet | 81.48 | 81.89 | 81.26 (1.39X) | 81.24 (7.72X) |
| Q8BERT | 80.28 | 80.99 | 80.14 (1.27X) | 80.11 (6.61X) |

| Network | Baseline | Accuracy Focus | Speed Focus (Speedup) | Size Focus (Compression) |
|---|---|---|---|---|
| BERT-Large | 12.04 | 11.92 | 12.08 (2.47X) | 12.09 (13.16X) |
| GPT-2 | 14.21 | 14.08 | 14.24 (2.08X) | 14.23 (7.64X) |

**Tuning the Approximation Knobs:** In this work, we considered a tight accuracy constaint of <0.5% accuracy degradation while optimizing the model, and determined the hyperparameter values (PruneThreshold and ApproxThreshold) empirically for that constraint. However, users of different applications and platforms may be willing relax the accuracy constraint for obtaining faster or smaller models. In view of this, we demonstrate the ability of our framework to operate at different points in the speed-size-accuracy tradeoff curve (Fig. 3) through different values of hyperparameters. We note that directly using optimized pre-trained Transformers for inference works best when there is a need

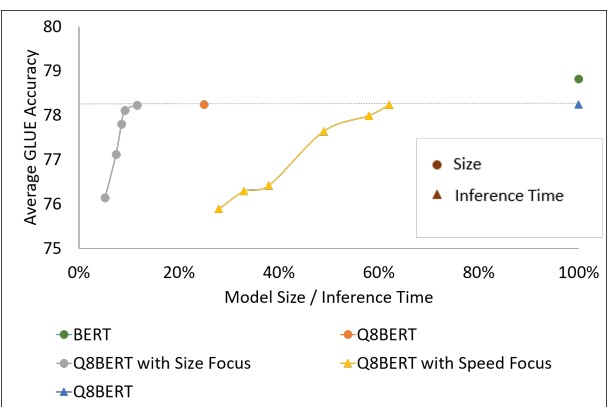

Figure 3: **Tuning the SA Approximation Knobs with Size and Speed Focus.** The average GLUE scores across the 9 tasks using Q8BERT are reported for different acceptable accuracy loss levels.

for significant speed/size improvement with negligible loss in accuracy (<2%), or if there is a need for more accurate models. When significant degradation in accuracy (>3%) is acceptable, techniques that distil knowledge into simpler networks that no longer maintain the structure of Transformers may be more beneficial. Even in these situations, our techniques are still useful, since they serve as better teachers/baselines during distillation/architecture search.

**Comparison to previously proposed compression techniques:** A majority of previous works for improving efficiency of Transformers directly pre-train efficient models from scratch. Using a representative subset of these networks (covering the most commonly used techniques used to create efficient models), we demonstrate that our techniques are complementary, since these efficient networks are still fine-tuned for different downstream tasks, providing opportunities for optimization. In addition, we show that our techniques are also complementary to Q8BERT, a post-training quantization method. Poor Man's BERT (Sajjad et al. (2020)) evaluated several layer dropping strategies that do not require pre-training, and found top-layer dropping to produce least accuracy degradation across tasks. Comparing our framework to top-layer dropping, we observe greater speedups/compression at iso-accuracy across all tasks and networks, and the largest benefits are observed on Q8BERT, where the use of quantization greatly reduces the resilience of the network, making it unsuitable for drastic changes such as dropping entire layers. However, by approaching the problem of improving inference efficiency in a hierarchical manner and with finer granularity, we are able to exploit redundancies in the model missed by a layer-only strategy, achieving greater benefits without significant loss in accuracy. In fact, in our experiments, we observe that starting on the layers as a first step leads to worse models than starting with blocks. We find that the effect of removing an ATTN block of relatively high significance may be masked by removing the FFN block of very low significance in the same layer (and vice-versa), leading to low significance for the entire layer. This has consequences further along in the process, since removing a high-significance block greatly reduces further opportunities for pruning and approximating the model. For experiments with Layerdrop

(Fan et al. (2020)), we experiment on RoBERTA (Liu et al. (2019)) using fairseq (Ott et al. (2019)) pre-trained with a layer drop rate of 0.5, and then drop every other layer at fine-tuning time. For QBERT, we directly use the results reported by the authors (Table 3).

Table 3: **Comparison to previously proposed compression techniques.** Results reported are averaged across the GLUE tasks, unless otherwise specified.

| Network | Pruning/ Optimization Strategy | Accuracy | Speedup | Compression |
|---|---|---|---|---|
| BERT-Base | None | 78.58 | 1x | 1x |
| Q8BERT | 8-bit Integer Quantization | 78.25 | 1x | 4x |
| | + Speed Focus | 78.23 | **1.61x** | 6.09x |
| | + Size Focus | 78.24 | 1.19x | **8.6x** |
| DistilBERT | Knowledge Distillation | 76.59 | 1.6x | 1.4x |
| | + Speed Focus | 76.57 | **2.26x** | 3.2x |
| | + Size Focus | 76.57 | 1.92x | **7.84x** |
| AlBERT | Parameter Sharing + Factorized Embeddings | 81.2 | 1.7x | 9.09x |
| | + Speed Focus | 81.03 | **2.35x** | 9.2x |
| | + Size Focus | 81.03 | 1.86x | **11.64x** |
| RoBERTa + LayerDrop − MNLI | Layer Dropping (6 Layers dropped) | 82.81 | 1.89x | 1.64x |
| | + Speed Focus | 82.79 | **2.33x** | 1.92x |
| | + Size Focus | 82.76 | 1.99x | **2.36x** |

| Network | Pruning/ Optimization Strategy | Accuracy | Speedup | Compression |
|---|---|---|---|---|
| Poor Man's BERT (applied to Q8BERT) | Top Layer Dropping (4 Layers dropped) | 75.15 | 1.39x | 1.24x |
| | Q8BERT + Speed Focus (iso-accuracy point) | 75.16 | **5.08x** | 8.68x |
| | Q8BERT + Size Focus (iso-accuracy point) | 75.15 | 2.92x | **10.94x** |
| QBERT − MNLI | Hessian-based mixed precision Quantization | 81.75 | 1x | 9.01x |
| | Q8BERT + Size Focus (iso-accuracy point) | 81.71 | 2.12x | 9.06x |

**Impact on fine-tuning time.** Unlike the baseline models, our framework requires multiple fine-tuning passes to optimize the model (Table 4). We minimize this overhead in two ways. First, since our iterative method potentially eliminates a component in each pass and our ordering of elements ensures that time-consuming components are eliminated early, each subsequent optimization fine-tuning pass takes less time. Second, for the optimization fine-tuning passes, we do not use the entire dataset for large datasets. Instead, we compute the thresholds

| Transformer | Task | Fine-tuning Passes for Optimization | Optimization Time (minutes) | Final Fine-tuning Time (minutes) | Total time (minutes) |
|---|---|---|---|---|---|
| BERT-Base | MNLI | 118 | 356 | 154 | 510 |
| BERT-Base | WNLI | 82 | 6.5 | 0.16 | 6.66 |
| DistilBERT-Base | MNLI | 44 | 93 | 94 | 187 |

Table 4: **Optimization and fine-tuning time.** The baseline final fine-tuning times are 189, 0.3 and 114 minutes respectively. For MNLI, we use only 15% of the training data for each pass during optimization. For WNLI (which has only 635 training samples), we use the entire training data.

based on a smaller subset of the target data. Specifically, we randomly sample a small subset of the training data (~20%) to fine-tune the model, and a validation set (~15% of the training set) to characterize significance. We find empirically that doing so results in the same elements getting pruned and approximated as when the entire training data is used. We further see that this subsampling is robust across models; if the reduced dataset works for one model, it works for all other models. Thus, by both greedily reducing the size of the model to be fine-tuned as well as reducing the amount of work performed for each optimization fine-tuning pass, we can quickly explore the search space.

## 6 CONCLUSION

We proposed an approximate computing framework to optimize pre-trained Transformers. The framework identifies elements that are insignificant for the downstream task at hand, and applies techniques to approximate these elements. We demonstrated that this framework can be adapted to produce models that are faster, smaller or more accurate, depending on the user's constraints. Using this framework, we produced models that were up to 4.22× faster, up to 14.08× smaller (with less than 0.5% relative accuracy degradation) and up to 5.46% absolute points more accurate with simultaneous speed and size benefits.

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

## A  DETAILED DESCRIPTION OF THE OPTIMIZATION FRAMEWORK

### A.1  ALGORITHMS AND ILLUSTRATION OF PRUNING STRATEGIES

---

**Algorithm 1:** Creating TransElement Queue

---

**Input:** Pre-trained Transformer T, Focus of optimization F, Downstream task D
**Output:** TransElement Queue Q, containing ordered elements of T for Significance Analysis
Q = $empty\_queue()$
**if** *FFN Blocks are more time-consuming/parameter-intensive* **then**
    **if** *Knowledge in bottom layers is more important* **then**
        **for** *layer = 1 to $num\_layers$* **do**
            Q[granularity=0].push($FFN\_block[layer]$)
            Q[granularity=1].push($FFN\_block[layer].Weight\_Groups$)
        **for** *layer = 1 to $num\_layers$* **do**
            Q[granularity=0].push($ATTN\_block[layer]$)
            Q[granularity=1].push($ATTN\_block[layer].Attention\_Heads$)
            Q[granularity=2].push($ATTN\_block[layer].Weight\_Groups$)
    **else if** *Knowledge in top layers is more important* **then**
        **for** *layer = $num\_layers$ to 1* **do**
            Q[granularity=0].push($FFN\_block[layer]$)
            Q[granularity=1].push($FFN\_block[layer].Weight\_Groups$)
        **for** *layer = $num\_layers$ to 1* **do**
            Q[granularity=0].push($ATTN\_block[layer]$
            Q[granularity=1].push($ATTN\_block[layer].Attention\_Heads$)
            Q[granularity=2].push($ATTN\_block[layer].Weight\_Groups$))
**else if** *ATTN Blocks are more time-consuming/parameter-intensive* **then**
    **if** *Knowledge in bottom layers is more important* **then**
        **for** *layer = 1 to $num\_layers$* **do**
            Q[granularity=0].push($ATTN\_block[layer]$)
            Q[granularity=1].push($ATTN\_block[layer].Attention\_Heads$)
            Q[granularity=2].push($ATTN\_block[layer].Weight\_Groups$))
        **for** *layer = 1 to $num\_layers$* **do**
            Q[granularity=0].push($FFN\_block[layer]$)
            Q[granularity=1].push($FFN\_block[layer].Weight\_Groups$)
    **else if** *Knowledge in top layers is more important* **then**
        **for** *layer = $num\_layers$ to 1* **do**
            Q[granularity=0].push($ATTN\_block[layer]$)
            Q[granularity=1].push($ATTN\_block[layer].Attention\_Heads$)
            Q[granularity=2].push($ATTN\_block[layer].Weight\_Groups$)
        **for** *layer = $num\_layers$ to 1* **do**
            Q[granularity=0].push($FFN\_block[layer]$)
            Q[granularity=1].push($FFN\_block[layer].Weight\_Groups$)
return $Q$

---

---

**Algorithm 2:** Significance Analysis

---

**Input:** Current state of the Transformer model T , Fine-tuning-dataset (or its reduced subset) D, TrialElement E, Thresholds {Pruning_Threshold, Approximation_Threshold}
**Output:** Action to be performed on E (whether to prune E, approximate E, or retain E as-is)
action = NULL
$T1 = \text{T} - \text{E}$
$TransElement\_Loss = Fine\_tune(\text{T1,D})$
**if** $TransElement\_Loss < Pruning\_Threshold$ **then**
  ⌊ action = "Prune"
**else if** $TransElement\_Loss >= Pruning\_Threshold\ and$
 $TransElement\_Loss < Approximation\_Threshold$ **then**
  ⌊ action = "Approximate"
return $action$

---

---

**Algorithm 3:** Transformer Optimization

---

**Input:** Pre-trained Transformer T , Fine-tuning-dataset D (and its reduced subset D' for large datasets, else D'=D), Focus of Optimization F, Acceptable Accuracy Loss A
**Output:** Optimized and fine-tuned Transformer for the given task
T', $Baseline\_Loss$ = Fine-tune(T,D')
$Pruning\_Threshold$, $Approximation\_Threshold =$
 $Compute\_Thresholds(Baseline\_Loss, F, A)$
Q = $Create\_TransElement\_Queue(T, F, D')$
granularity = 0
**while** *Q is not empty* **do**
  | TrialElement = Q[granularity].pop()
  | action =
  | $Significance\_Analysis[T, D', TrialElement, Pruning\_Threshold, Approximation\_Threshold]$
  | **if** *action = "Prune"* **then**
  |   | $modified\_TransElement$ = None
  |   ⌊ Q.pop(All elements encompassed by TrialElement)
  | **else if** *action = "Approximate"* **then**
  |   | **if** *granularity = max_granularity* **then**
  |   |   | **if** *Focus = "Accuracy"* **then**
  |   |   |   ⌊ $modified\_TransElement$ = trialElement
  |   |   | **else if** *Focus = "Size"* **then**
  |   |   |   ⌊ $modified\_TransElement = quantize\_lower$(trialElement)
  |   |   | **else if** *Focus = "Speed* **then**
  |   |   |   | **if** *TrialElement is encompassed by a FFN block* **then**
  |   |   |   |   ⌊ $modified\_TransElement$ = trialElement
  |   |   |   | **else if** *TrialElement is encompassed by an ATTN block* **then**
  |   |   |   |   ⌊ $modified\_TransElement = Sign\_Matching$(trialElement)
  |   | **else**
  |   |   ⌊ $modified\_TransElement$ = TrialElement
  | **else**
  |   | Q.pop(All elements encompassed by TrialElement)
  |   ⌊ $modified\_TransElement$ = TrialElement
  | **if** *Q[granularity] is empty* **then**
  |   ⌊ granularity++
  ⌊ T = T - TrialElement + $modified\_TransElement$
T, $Final\_Loss = Fine\_tune(T, D)$
return T

---

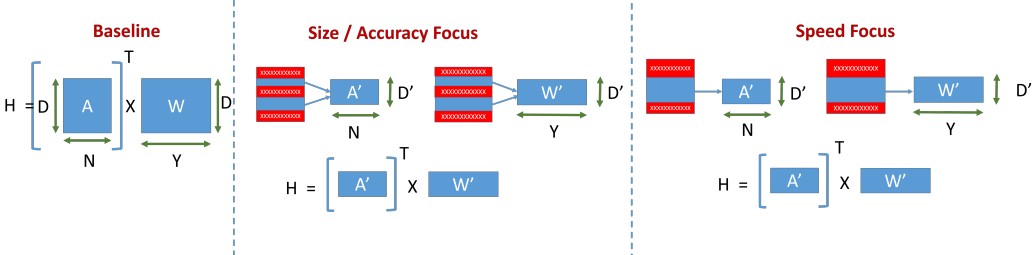

Figure 4: **Illustration of pruning techniques used in FFN Blocks.** 'A' denotes activations and 'W' denotes weights, transposed for clarity. When optimizing for accuracy/size, unstructured pruning is used. When optimizing for speed, our greedy shrinking method is used to prune weight groups, so that the remaining weight groups form a contiguous dense block.

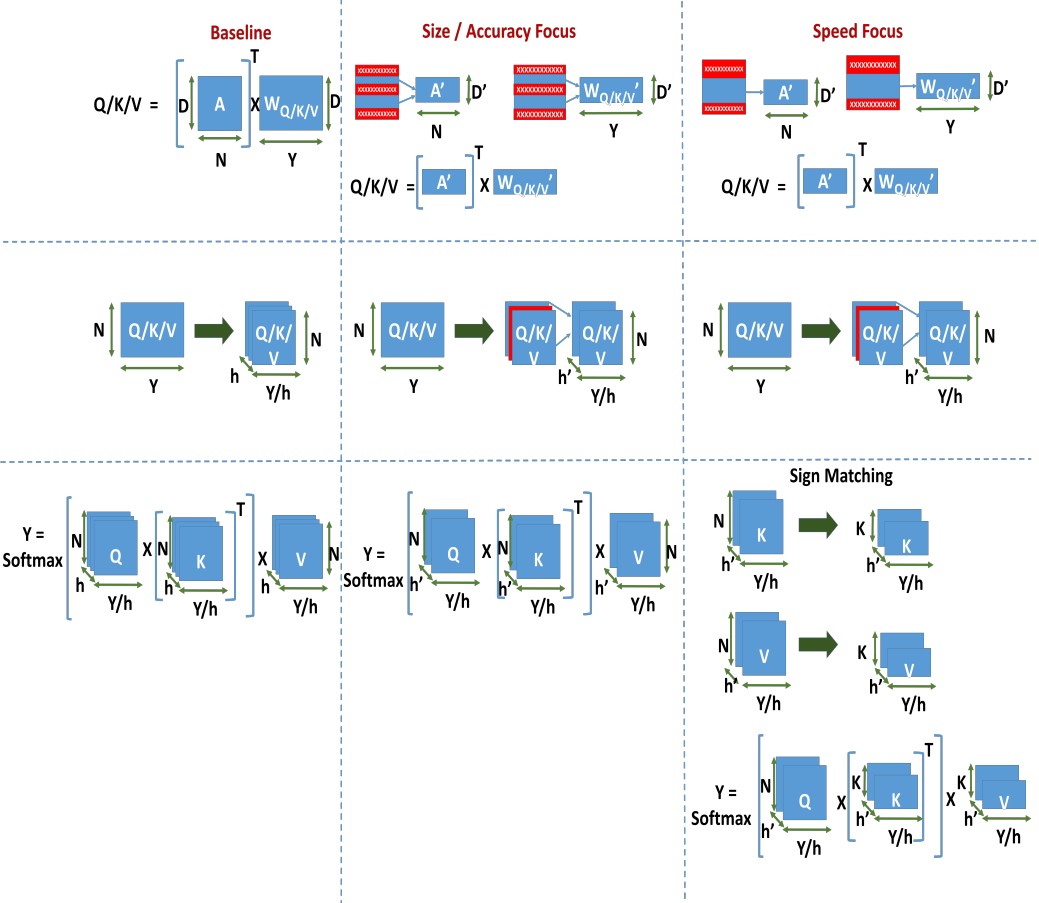

Figure 5: **Illustration of pruning techniques used in ATTN Blocks.** 'A' denotes activations and 'W' denotes weights, transposed for clarity. Pruning heads in the second step is illustrated this way for the sake of clarity. In the actual implementation, heads are pruned by pruning the corresponding weight groups along the $y$ dimension of $W_{Q/K/V}$ in step 1. Greedy shrinking is not used when optimizing for speed for pruning heads.

## B   SIGN MATCHING - DETAILED DESCRIPTION AND ABLATION STUDIES

### B.1   ALGORITHM

---
**Algorithm 4:** Sign Matching
---
**Input:**  Set of query vectors Query = $[q_1, q_2, ..., q_n]$, set of key vectors Key = $[k_1, k_2, ..., k_n]$,
     set of value vectors Value = $[v_1, v_2, ..., v_n]$, number of key vectors to select K
**Output:** Set of key vectors with highest sign match with query vectors Key' = $[k'_1, k'_2, ..., k'_K]$ ,
     set of corresponding value vectors Value' = $[v'_1, v'_2, ..., v'_K]$

```
/* Build sign representation of Query vectors                        */
i ← 1
count ← 0
while i <= d do
    j ← 1
    while j <= n do
        if q_{j,i} > 0 then
            count[i] ← count[i] + 1
        j ← j + 1
    if count[i] >= (n/2) then
        val[i] ← 1
    else
        val[i] ← −1
    i ← i + 1
/* Compute sign matches of Key vectors with the representative
   Query vector                                                      */
i ← 1
matches ← 0
while i <= n do
    H_Dist[i] ← Hamming_Distance(sign(k_i), val)
    i ← i + 1
matches ← indices(sort_ascending(H_Dist))
matches ← matches[1 : K]
K' ← gather(K(matches))
V' ← gather(V(matches))
```
---

### B.2   SIGN MATCHING IN AUTO-REGRESSIVE MODELS

In Auto-regressive models (XL-Net, GPT-2, Transformer-XL, etc.), tokens are not allowed to attend to tokens in the future, and an attention mask is applied to set the corresponding weights to a large negative value. This is a problem because the key vectors selected by SM may be such that vectors at the start of the sequence (first few query vectors) may not be able to attend to any of the key vectors (i.e., their attention outputs will be 0), leading to significant loss of information and degradation in accuracy. We avoid this by selecting the top-scoring $(K/4)$ vectors from the top quarter of the key matrix, and the top-scoring $(3K/4)$ vectors from the overall key matrix and not included in the $(K/4)$ vectors initially selected, instead of directly selecting top $K$ vectors from overall key matrix. This reduces the probability of vectors having no vectors from their past to attend to.

### B.3   COMPARISON TO OTHER DYNAMIC KEY-SELECTION TECHNIQUES

We compare our Sign-Matching based attention with other intuitive dynamic Key-selection that do not require training from scratch and provide speedups even at small context lengths. We find that Sign Matching provides the best trade-off between accuracy and speed. The techniques considered are described below:

**Norm-based Selection (NBS).** NBS is based on the idea that the "most important" key vectors (corresponding to the "most important" tokens in the dataset) will have highest norm, and hence

highest attention (dot-product) with the query vectors. The key vectors are ranked in descending order of their norm, and the top $K$ vectors are selected. Attention is then computed only between the query vectors and the selected key vectors.

**Value-based Selection (VBS).** One of the disadvantages of NBS is the fact that a vector with only one very large value will have high norm, but it is unlikely to produce high dot-products with other vectors. VBS overcomes this by using distribution of values in a vector, rather than the norm, as the selection criteria. In particular, we count the number of elements in each vector greater than a specified "value threshold". We then select the $K$ vectors with maximum number of elements with absolute values greater than the "value threshold".

**Grouped Attention (GA).** GA places vectors that are likely to have high dot-product with each other in the same group, and vectors likely to have low dot-product with each other in different groups with high probability. The concept of GA was previously explored in Reformer (Kitaev et al. (2020)), where Locality Sensitive Hashing was used to group vectors. However, since we apply this approximation only to resilient layers, we use a simpler and faster grouping criterion, the position of maximum and minimum value. In addition, there is no need for multiple iterations of grouping and computing attention scores that was used in Reformer to ensure that query-key pairs with high attention scores were placed in the same group in at least one of the iterations. Both of these factors together greatly reduce our overheads, enabling speedups even at small context lengths. Our grouping criteria is based on the intuition that vectors that have highest positive/negative values in the same position will have high dot-products with each other. Attention scores are then computed only between query-key pairs in the same group, since they are most likely to have high dot-products with each other. We limit the number of key vectors in each group to $K$ vectors with the highest absolute value in that position, and therefore GA scales linearly in time and memory with context length instead of quadratically.

### B.4 Speedup with increasing context length

Since Sign Matching is a linear-time approximation of the quadratic self-attention operation, speedup increases significantly with increase in context length. As context length increases, ATTN Blocks become time-dominant, and hence more emphasis is placed on these blocks by our framework. In addition, the memory requirements increase quadratically with context length due to the self-attention operation, making Transformers extremely memory-bottlenecked. Sign Matching helps alleviate this bottleneck. Through the combination of these factors, we find a large increase in speedup as context length increases from our Sign Matching technique (Table 5).

Table 5: **[Left] Comparing Sign-Matching with other dynamic Key-selection algorithms.** Reduction in accuracy and time are reported on MRPC using BERT-Base (context length of 128). They are measured as the difference in accuracy and inference time when all approximable ATTN Blocks are replaced with approximate versions of the self-attention operation. **[Right] Speedup from Sign Matching at increased context lengths.**

| Approximation | Reduction in Accuracy | Reduction in time |
|---|---|---|
| Norm-Based Selection | 0.22% | 31% |
| Value-Based Selection | 0.14% | 32% |
| Sign Matching | 0.06% | 32% |
| Grouped Attention | 0.05% | 6% |

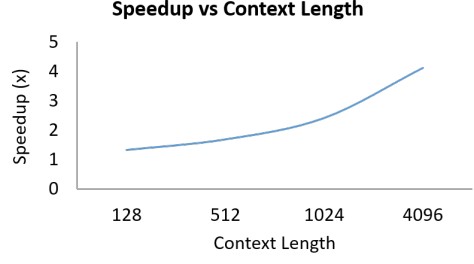

## C   Analysis of the optimization framework

### C.1   Previously proposed methods for estimating importance of elements are not applicable

Due to the enormous size of Transformer models, brute-force approaches to estimating importance are not feasible. In addition, previously proposed techniques for efficient importance estimation are not well-suited due to the fact that Transformers cannot be repeatedly fine-tuned to recover the accuracy losses from approximating the model, since they very quickly overfit the training data for the downstream tasks (usually within 5 epochs). Therefore, Taylor expansion, which uses gradient of loss to estimate importance, is not reliable, as evidenced in Table 6. We observe that in addition to providing greater control over the accuracy of the final

| Method | Accuracy | Model Speedup | Model Compression |
|---|---|---|---|
| Baseline | 84.06 | 1x | 1x |
| Taylor Expansion (negative gradient of loss) | 80.65 | 2.64x | 4.53x |
| **Ours (Iso-Accuracy point)** | **80.66** | **3.28x** | **6.06x** |
| Taylor Expansion (smallest absolute gradient of loss) | 80.69 | 2.51x | 4.48x |
| **Ours (Iso-Accuracy point)** | **80.7** | **3.3x** | **6.09x** |
| **Ours (Accuracy Focus)** | **84.63** | **1.14x** | **1.98x** |

Table 6: **Comparison of our method of estimating importance with previously proposed methods.** We show results on MRPC using DistilBERT with Accuracy Focus.

model (and the ability to increase accuracy), our framework also provides better speedup and compression at similar accuracy.

### C.2   Evaluation of heuristics used

We compare different possible heuristics to the ones used in our framework (Table 7) on MRPC using DistilBERT. When we remove the element with the lowest loss in each iteration (with loss characterized using our method), there is negligible change in the quality of the final model produced, but the fine-tuning+optimization process is an order of magnitude slower if elements are still considered in the order of coarser to finer granularity, and two orders of magnitude slower otherwise compared to our approach. If the loss is characterized using Taylor expansion, it greatly destabilizes the system, leading to models that do not meet the accuracy constraints. To drive home the fact that our greedy approach combined with a global error bound does not lead to inferior models, we use an adaptive loss threshold. In particular, we use a very tight constraint when analyzing elements at coarse granularities, and relax the constraint as we move towards finer granularities. We again find that there is negligible change in the quality of the final model produced, but the fine-tuning+optimization process is significantly slower. We hypothesize that a single global error bound is sufficient because we order the elements in such a way that for the given task at hand, we intuitively expect that the elements at the head of the queue are likely to be removed using the linguistic knowledge in different layers. Therefore, it is reasonable to expect that if an element at the head of the queue is identified by our framework as prunable, it can be pruned without using up a large portion of our error budget.

### C.3   Gains from different optimization techniques

The gains obtained from different optimization techniques for different tasks and models depends on two factors: Number of elements to which each technique has been applied, and the gain from applying each technique to a single element. In general, we observe that the largest gains are obtained from pruning entire blocks that are most time-consuming/ parameter-intensive. This means that at small context lengths (such as BERT-Base on MRPC in Fig. 6), pruning entire FFN Blocks produces maximum gain. And at large context lengths (such as GPT-2 Base on Penn Treebank in Fig. 6), pruning entire ATTN Blocks provides maximum gain. Our analysis also demonstrates that all techniques are vital for producing highly optimized models, since no single strategy can provide drastic gains with minimal accuracy degradation.

Table 7: **Comparison of different heuristics. 1** denotes a strategy where it examines all elements (of the same granularity), and then prunes the element causing lowest loss. If marked as Taylor, Taylor expansion is used to estimate importance with a single pass of the validation set. Otherwise, importance is estimated by our method with multiple fine-tuning passes before an element is pruned. **2** denotes a greedy-strategy where if an element is found to maintain accuracy within the respective thresholds when removed, it is pruned/approximated right away. **a** denotes hierarchical processing of elements. **b** denotes non-hierarchical processing of elements (all elements are considered to be of same granularity for the purpose of importance estimation).

| Heuristic | Optimized Model Accuracy | Optimized Model Compression | Optimized Model Speedup | Optimization Time (minutes) |
|---|---|---|---|---|
| Drop element with lowest loss in each iteration - Taylor[1a] | 83.22 | 2.64x | 2.03x | 61 |
| Drop element with lowest loss in each iteration[1a] | 84.05 | 1.95x | 1.36x | 1081 |
| Random[2a] | 84.06 | 1.92x | 1.34x | 66 |
| Non-hierarchial (Min Loss)[1b] | 84.03 | 2.01x | 1.37x | 13822 |
| Non-hierarchial (Greedy)[2b] | 84.03 | 2.01x | 1.37x | 3634 |
| Adaptive Threshold[2a] | 84.05 | 1.95x | 1.36x | 143 |
| TransElement Ordered Queue (Ours)[2a] | 84.06 | 1.92x | 1.34x | 39 |

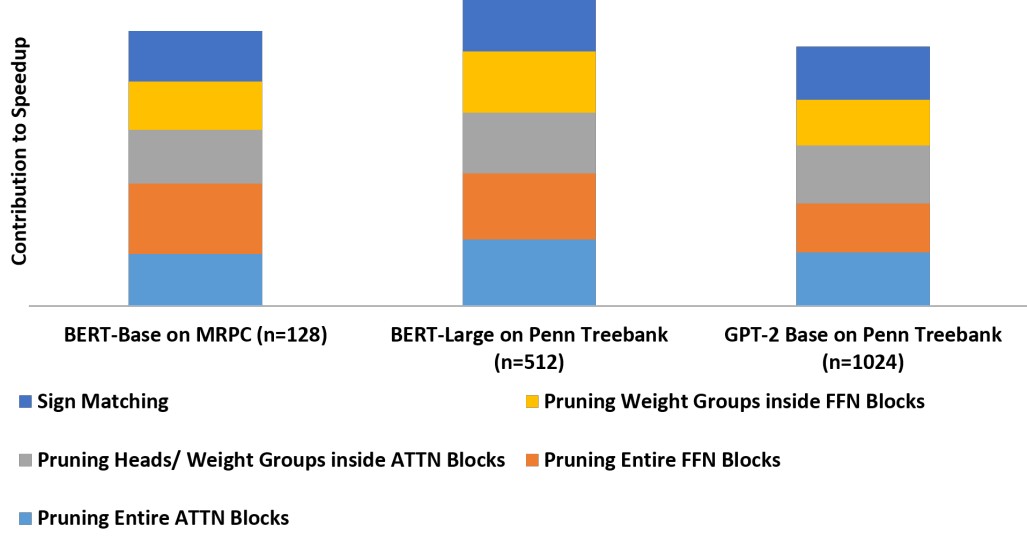

Figure 6: **Gains from different optimization techniques using Speed Focus.**

# D ANALYSIS OF IMPORTANT ELEMENTS ACROSS DOWNSTREAM TASKS AND TRANSFORMER MODELS

We study which blocks are pruned and approximated for different downstream tasks using different Transformers (Fig. 7). We find that the differences in importance of blocks are more pronounced across different tasks than across different models. For example, for sentiment analysis, long-range dependency information is not very important. Hence, for all models fine-tuned for sentiment analysis, we observe that components in later layers (closer to the output) are more likely to be pruned. Across models, we only observe subtle differences. For example, we find that XLNet(auto-regressive) is able to learn important task-specific information earlier than BERT(non auto-regressive), similar to the observation made in (Sajjad et al. (2020)). Hence, we are able to drop more components (in earlier layers) in XLNet than in BERT, leading to more efficient models for inference. In DistilBERT (a distilled model), we find that there is a clear demarcation in linguistic knowledge across layers due to the reduced capacity of the model. This is evidenced by the fact that components in the top four layers are never pruned across all Language Understanding tasks, while the boundaries are more soft in the original models. At a high level, these trends agree with previous works on understanding how Transformers process language (Jawahar et al. (2019)) that use probing classifiers to discern the linguistic knowledge captured by the different layers. For example, on GLUE tasks, we expect that long-range dependency information is not required for most tasks, since most of these tasks depend on local contexts. This is confirmed by the fact that blocks in layers are more likely to be pruned/approximated than earlier layers using our framework. Similarly, we expect that this is not the case for Language Modelling, since long-range dependency information is vital to fully understand the text. This is also confirmed by the observed trends using our framework. As future work, our framework can be combined with previously proposed techniques to gain deeper understanding of the working of Transformers, especially at finer levels of granularity.

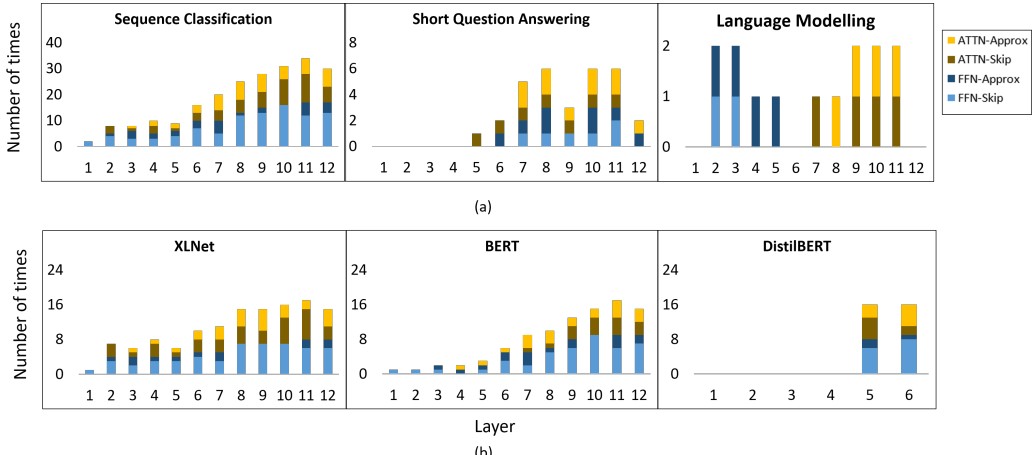

Figure 7: **Distribution of unimportant blocks across (a) downstream tasks and (b) Transformers.**

# E   EXPERIMENT DETAILS AND HYPERPARAMETERS

## E.1   DESCRIPTION OF TASKS AND TRANSFORMERS USED IN OUR EXPERIMENTS

Table 8: **[Left] Transformer models and [Right] downstream tasks used in our experiments and studies.**

| Transformer | Layers | Parameters(M) | Auto-Regressive? | Optimizations |
|---|---|---|---|---|
| BERT-Base | 12 | 110 | No | None |
| BERT-Large | 24 | 340 | No | None |
| AlBERT-Base | 12 | 12 | No | • Uses factorized embeddings and cross-layer parameter sharing
• Faster and has less parameters than BERT. |
| DistilBERT-Base | 6 | 66 | No | • Uses Knowledge Distillation in the pre-training phase
• Has half the number of layers with <3% accuracy loss compared to BERT. |
| Q8BERT-Base | 12 | 110 | No | • Quantizes BERT to 8-bit integer weights and activations
• Uses Fake Quantization and Quantization Aware Training in the fine-tuning phase.
• 4x smaller than BERT with potential for greater speedups on optimized int8 kernels with <1% accuracy loss. |
| XLNET-Base | 12 | 110 | Yes | None |
| GPT-2 Base | 12 | 117 | Yes | None |

| Task | Dataset | Transformers used | Context Length |
|---|---|---|---|
| General Language Understanding | GLUE | Q8BERT-Base, DistilBERT-Base, XLNET-Base | 128 |
| Question Answering | SQUAD v1.1 | AlBERT-Base | 384 |
| Language Modelling | Penn Treebank | BERT-Large, GPT-2 Base | 512(BERT-Large)/1024 (GPT-2) |

## E.2   HYPERPARAMETERS USED IN OUR EXPERIMENTS

Table 9: **Hyperparameters used in our experiments.**

| Hyperparameter | Accuracy Focus | Speed Focus | Size Focus |
|---|---|---|---|
| Fine-tuning Epochs (baseline) | 3 | 3 | 3 |
| Fine-tuning Epochs (approx) | 3 | 3 | 3 |
| Learning rate | 2.00E-05 | 2.00E-05 | 2.00E-05 |
| Warmup steps | 0 | 0 | 0 |
| Batch size | 32 (fine-tuning) / 8 (inference) | 32 (fine-tuning) / 8 (inference) | 32 (fine-tuning) / 8 (inference) |
| *W* (size of one weight group in SA) | 256 | 256 | 256 |
| *K* (number of key elements selected in Sign Matching) | N/A | 16, if n<=128
64, if 128<n<1024
128, if n>=1024 | N/A |
| *Skip_Threshold* (<0.5% accuracy loss) | min_loss | 1.005*baseline_loss | 1.005*baseline_loss |
| *Approx_Threshold* (<0.5% accuracy loss) | 1.02*min_loss | 1.02*baseline_loss | 1.02*baseline_loss |

| TransElement Loss | Number of bits used to represent the TransElement |
|---|---|
| <1.005*baseline | 0 |
| 1.005*baseline_loss - 1.006*baseline_loss | 4 |
| 1.006*baseline_loss - 1.007*baseline_loss | 5 |
| 1.007*baseline_loss - 1.008*baseline_loss | 6 |
| 1.008*baseline_loss - 1.009*baseline_loss | 7 |
| >1.009*baseline_loss | 8 |

- min_loss refers to the smallest loss seen during the Significance Analysis process. We use this as the threshold when optimizing for accuracy since the goal is to produce a model with the smallest possible loss.

- baseline_loss refers to the loss when the pre-trained transformer is used as-is, without any TransElements skipped or approximated. We use this to compute the threshold when optimizing for speed and size since small accuracy loss is acceptable.

- Table 8 [Right] shows how we quantize insignificant TransElements to lower precisions when operating with Size ApproxFocus.

# F    RESULTS ON THE GLUE TEST SET

While our main results are on the dev set following standard practice, we report results on the test set also using the BERT (base) model in Table 10. We use the GLUE evaluation server to obtain the scores, and make use of code from Xu et al. (2020) to prepare the data for submission.

Table 10: **Results on GLUE Test Set.** We report Matthews correlation for CoLA, Pearson Correlation for STS-B and accuracy for all other tasks, averaged across 10 random seeds. We report only "matched" accuracy for MNLI.

|  | CoLA | MNLI | MRPC | QNLI | QQP | RTE | SST-2 | STS-B | WNLI | Average |
|---|---|---|---|---|---|---|---|---|---|---|
| **Baseline** | 50.38 | 84.37 | 82.11 | 89.74 | 88.76 | 61.65 | 94.94 | 82.8 | 58.19 | 76.99 |
| **Accuracy Focus** | 51.21 | 84.59 | 83.15 | 89.8 | 89.44 | 63.82 | 95.02 | 83.17 | 65.14 | 78.37 |
| **Speed Focus (Speedup)** | 50.42 (2.81x) | 84.21 (1.35x) | 81.97 (2.12x) | 89.58 (1.32x) | 88.47 (1.98x) | 61.32 (3.03x) | 94.68 (1.37x) | 82.62 (1.96x) | 59.33 (3.49x) | 76.95 (2.16x) |
| **Size Focus (Compression)** | 50.18 (13.13x) | 84.03 (5.12x) | 81.72 (6.28x) | 89.43 (5.16x) | 88.24 (8.08x) | 61.4 (10.9x) | 94.39 (5.58x) | 82.42 (7.12x) | 59.13 (14.14x) | 76.77 (8.39x) |

