# OpenReview forum: "Optimizing Transformers with Approximate Computing for Faster, Smaller and more Accurate NLP Models"
_ICLR.cc/2021/Conference — Reject_

### Official Review · AnonReviewer4 · 2020-10-27
**Techniques need more justification and presentation needs polishing**

**Rating:** 4
**Confidence:** 5

**Review:**

Thanks to the authors for the detailed feedback! I still have concerns about the clarity of the presentation, and some contributions of the papers are not strong enough, so I'll keep my score.

===

Summary:

This paper presents a framework to systematically perform pruning and layer approximation. The framework includes a queue of potential elements for compression. At each time step, the framework will evaluate the head element of the queue, try to prune the whole element or perform approximation (quantizing, pruning attention heads, and approximating with sign-matching attention), and keep the transformation only if the loss in performance is acceptable. The paper performs experiments with various models on GLUE and shows speedups or compressions compared to the original model.

Reasons for score:

The techniques used in the paper are not novel, and the choices on how to apply multiple compression techniques need more justification. The experiment results are okay but not surprising. The presentation of the paper needs to be polished. See below for more details.

Pros:

1. I like the insight that {approximate, fine-tune, approximate} cycles doesn’t work for fine-tuning.
2. I like the insights used to determine which elements to be examined first: start from the larger blocks and later layers. I hope this point can be emphasized more and compared with more brute-force and less-efficient algorithms. For example, for each round, one can choose a layer that causes the least loss of performance to prune. You can compare your greedy algorithm with this algorithm to show the gain of using a less efficient algorithm is not significant.
3. The sign-matching attention proposed in the paper is new. I would like to see more emphasis and ablation studies on the effectiveness of this special module.

Cons:

1. It is well-known that compressing the model is easier during the fine-tuning phase [1, 2]. I don’t think this should be a contribution to emphasize for the paper.
2. The whole compression framework has a single global error bound. Combining this with the greedy layer-by-layer approach taken by the framework, will the following case be possible: a layer that is early in the queue causes a huge drop of accuracy and thus makes all the future layers impossible to remove because the global error bound has been reached. A better way is to only remove the layer with the lowest loss reduction. It will be better to justify this point with an ablation study, or at least show the final pruned model doesn’t have this issue in the paper.
3. At the end of page 5: “When optimizing for speed, however, removing weight groups with low significance from arbitrary locations does not help, since it introduces unstructured sparsity in the weight matrix that can be difficult to exploit to achieve speedups.” It’s true that if you remove random entries in a matrix will not help for the actual speedups, but you can remove an arbitrary set of rows of the matrix, and then restructure the weight matrix (i.e. concatenate all the remaining rows to form a new matrix) to make it efficient for modern parallel hardwares.
4. I don’t really understand the point of using accuracy as the final goal. If the framework is for compression, the goal should be about speedup or size. If accuracy really matters, it should be enforced as the threshold instead of as the final goal. Also, I don’t see the difference in the framework between using speedup or size as the goal, since all the thresholds are defined by loss.
5. The results in the paper are okay, but compared to previous works in computer vision [3], it seems that the model size can be further compressed.
6. There are multiple places where the presentation can be improved:

    a. It’s more clear to use a pseudo-code instead of a diagram in Figure 2.

    b. It should be more clear to present Table 1 as multiple tables.

    c. It’s better to put the results comparing with previous works in a table (in the middle of page 8).

Minor comments:

- On page 5, 3rd paragraph from the bottom, “less the minimum loss” -> “less than minimum loss”

References:

[1] Jiao, Xiaoqi, et al. "Tinybert: Distilling bert for natural language understanding." arXiv preprint arXiv:1909.10351 (2019).

[2] Shen, Sheng, et al. "Q-BERT: Hessian Based Ultra Low Precision Quantization of BERT." AAAI. 2020.

[3] Han, Song, Huizi Mao, and William J. Dally. "Deep compression: Compressing deep neural networks with pruning, trained quantization and huffman coding." arXiv preprint arXiv:1510.00149 (2015).

---

> ### Author Response · Authors · 2020-11-20
> **Response to Reviewer4 (Part 1/3)**
>
> We thank the reviewer for their comments. We have addressed the individual concerns below.
> __1. Re: It is well-known that compressing the model is easier during the fine-tuning phase [1, 2]. I don’t think this should be a contribution to emphasize for the paper.__ \
> We thank the reviewer for pointing this out. We did not intend to emphasize this as a contribution. However, we do believe that it is important to point out that our techniques do not require training from scratch, since the computational demands of pre-training transformers are enormous (orders of magnitude higher than for other networks). Also, a vast majority of the works that aim to improve the efficiency of transformers do require training from scratch, and we note them in our related works section. We have modified the language in the revised manuscript to ensure that we are not emphasizing this as a contribution.\
> __2. Re: The whole compression framework has a single global error bound. Combining this with the greedy layer-by-layer approach taken by the framework, will the following case be possible: a layer that is early in the queue causes a huge drop of accuracy and thus makes all the future layers impossible to remove because the global error bound has been reached. A better way is to only remove the layer with the lowest loss reduction. It will be better to justify this point with an ablation study, or at least show the final pruned model doesn’t have this issue in the paper.__ \
> We have added an ablation study to appendix C in the latest version to address this question. In summary, when we remove the element with the lowest loss in each iteration (with loss characterized using our method), there is negligible change in the quality of the final model produced, but the fine-tuning+optimization process is an order of magnitude slower if elements are still considered in the order of coarser to finer granularity, and two orders of magnitude slower otherwise. If the loss is characterized using Taylor expansion, it greatly destabilizes the system, leading to models that do not meet the accuracy constraints. This is because Taylor expansion uses gradients of loss to estimate significance, which is not reliable when additional fine-tuning is not allowed to recover the accuracy loss caused by removing the element.\
> To further drive home this point, we use an adaptive loss threshold. In particular, we use a very tight constraint when analyzing elements at coarse granularities, and relax the constraint as we move towards finer granularities. We again find that there is negligible change in the quality of the final model produced, but the fine-tuning+optimization process is significantly slower. We hypothesize that a single global error bound is sufficient because we order the elements (enqueue) in such a way that for the given task at hand, we intuitively expect that the elements at the head of the queue are likely to be removed. Previous works have shown that different transformer layers capture distinct aspects of linguistic knowledge, and for each task, we order elements based on whether the knowledge captured by the element is important for the task or not. Therefore, it is reasonable to expect that if an element at the head of the queue is identified by our framework as prunable, it can be pruned without using up a large portion of our error budget.
> We realize that this fact was not adequately addressed, and it is not intuitively clear to readers why this would work. We thank the reviewer for highlighting this, and we have addressed this in the latest version.\
> __3. Re: At the end of page 5: “When optimizing for speed, however, removing weight groups with low significance from arbitrary locations does not help, since it introduces unstructured sparsity in the weight matrix that can be difficult to exploit to achieve speedups.” It’s true that if you remove random entries in a matrix will not help for the actual speedups, but you can remove an arbitrary set of rows of the matrix, and then restructure the weight matrix (i.e. concatenate all the remaining rows to form a new matrix) to make it efficient for modern parallel hardwares.__ \
> Yes, we agree that the weight matrix can be restructured. However, we need to dynamically restructure the activations also to match the restructured weight matrix, in order for the computations to be correct. In our experiments, we observed that the overheads of restructuring the activations (gathering values from non-contiguous locations, and putting them together) outweigh the benefits of the faster matrix-multiply operation. In addition, due to the presence of skip connections between the different transformer blocks, we cannot change the shape of the activations before-hand to match the shape of the new weight matrix, which could have potentially alleviated this overhead.

---

> > ### Author Response · Authors · 2020-11-20
> > **Response to Reviewer4 (Part 2/3)**
> >
> > __4. Re: I don’t really understand the point of using accuracy as the final goal. If the framework is for compression, the goal should be about speedup or size. If accuracy really matter, it should be enforced as the threshold instead of as the final goal. Also, I don’t see the difference in the framework between using speedup or size as the goal, since all the thresholds are defined by loss.__ \
> > We acknowledge the fact that pruning usually leads to a drop in accuracy, and the techniques proposed in literature aim to mitigate this accuracy loss. Therefore, when we proposed a pruning framework that actually improves accuracy, we needed to provide justification for its utility. In summary, we believe that existing compression frameworks are attractive only to users with resource constraints, since they do cause accuracy degradation. By providing the option of optimizing for accuracy, we believe that our framework will be of interest to all users, since it can produce models that are more accurate while also being smaller and faster. This is made possible by the unique characteristics of transformers, where over-parameterized transformers are first pre-trained on a difficult initial task, and then fine-tuned for other (often simpler) NLP tasks (with state-of-the-art performance in these tasks). During pre-training, different transformer layers capture different aspects of linguistic knowledge, and during fine-tuning, some of these aspects may be useless (or even harmful) for the task at hand. This is also why we propose an “optimization” framework that is useful for more users than a “compression” framework.\
> > We agree that all thresholds are defined by loss when optimizing for speed or size. However, the set of pruning and approximation techniques are different for the two cases. When optimizing for size, the goal is to produce models that are as small as possible while staying within the accuracy constraint. To this end, we use unstructured pruning and quantization to reduce model size as much as possible. Similarly, when optimizing for speed, the goal is to produce models that are as fast as possible while staying within the accuracy constraint. Hence, we use hardware-friendly structured pruning (using our “greedy shrinking” algorithm) and sign-matching based approximate attention. \
> > __5. Re: The results in the paper are okay, but compared to previous works in computer vision [3], it seems that the model size can be further compressed.__ \
> > We believe that there is a fundamental difference between transformers and conventional CNNs that limits compression: we cannot repeatedly prune a set of parameters and then fine-tune to regain accuracy. Previous works in computer vision do not have this limitation, and even in [3], fine-tuning is done to regain lost accuracy. In addition, following standard practice, we mainly demonstrate the effectiveness of our methods on base versions of the transformers, due to the prohibitive computational requirements of working with large state-of-the-art models like GPT-3(175B parameters). Based on our observations of scaling to larger models -- DistilBERT (66M parameters) can be compressed by 5.2x, BERT-base (110M parameters) can be compressed by 8.5x and BERT-Large (335M parameters) can be compressed by 13.7x with <0.5% accuracy degradation averaged across the 9 GLUE tasks -- we believe that as models become larger and larger, the compression rates using our framework will increase.\
> > __6. Re: There are multiple places where the presentation can be improved: a. It’s more clear to use a pseudo-code instead of a diagram in Figure 2. b. It should be more clear to present Table 1 as multiple tables. c. It’s better to put the results comparing with previous works in a table (in the middle of page 8).__ \
> > We thank the reviewer for bringing these points to our attention. We have addressed all of these points in our latest version. In particular, we have added detailed descriptions of the different strategies, including flowcharts and pseudocode, in Appendix A, in order to make them easily understandable without exceeding the page limit. We hope this makes the process clear and reproducible. We have also split up the tables (into Table 1 and Table 2) and better represented the comparison to previous works (Table 3).

---

> > > ### Author Response · Authors · 2020-11-20
> > > **Response to Reviewer4 (Part 3/3)**
> > >
> > > __7. Re: The sign-matching attention proposed in the paper is new. I would like to see more emphasis and ablation studies on the effectiveness of this special module__ \
> > > We thank the reviewer for highlighting this contribution. Sign Matching is a linear-time approximation of self-attention that does not require training from scratch, and provides significant speedup even when small context lengths (which are sufficient for many practical applications, including most tasks studied in this paper) are used. We have added an ablation study to Appendix B in the paper, comparing to other possible heuristics for dynamic Key-selection. We have also added a study on the benefits of Sign Matching at larger context lengths.

---

### Official Review · AnonReviewer2 · 2020-10-28
**Respectable engineering effort with promising experimental results**

**Rating:** 7
**Confidence:** 4

**Review:**

This paper presents a method for improving a fine-turned Transformer in terms of a specific metric such as size, speed, or accuracy. The candidates of removed elements are considered hierarchically with some heuristics and are evaluated in terms of training and validation loss to determine whether they should actually be removed from the model. The authors apply their method to several state-of-the-art Transformer models and show that they can produce fast and compact models without losing much accuracy.

Although the individual techniques employed to realize the whole pruning process are not particularly novel, the paper presents a well-thought-out approach to combine those and reports very promising experimental results. I think this is a nice contribution to the community, given that the computation cost is increasingly important in dealing with BERT-like models.

It seems to me that the authors used transformers whose weights are shared between different layers like Universal Transformers or ALBERT. Maybe I missed something, but I think the authors should clarify if this is really the case in the manuscript.

The entire process of pruning is a bit vague and hard to replicate. Would it be possible to describe the whole process in pseudo code? (Is Algorithm 1 the whole process?)

I think the authors should also describe the computational cost (or maybe wallclock time) required to perform the proposed pruning processes. It seems to me that the search space is rather large and requires a considerable amount of computation.

> p.5 … we prune the element only if the training/validation loss

I think you should be more specific here. How did you actually use both the training and validation loss?  Why do you need to look at the training loss when you are interested in the generalization error?

> p.5 … weight groups of (Wn) …

Why is this Wn? I thought this should be W.

Minor comments:
p.5 less the -> less than the?
p.6 doesn’t -> does not
p.6 ’’attention -> ``attention
p.7 second order -> second-order?

---

> ### Author Response · Authors · 2020-11-20
> **Response to Reviewer2**
>
> We thank the reviewer for their comments and positive review. We have addressed the individual concerns below.\
> __1. Re: It seems to me that the authors used transformers whose weights are shared between different layers like Universal Transformers or ALBERT. Maybe I missed something, but I think the authors should clarify if this is really the case in the manuscript.__ \
> We experiment with both kinds of models - transformers with weight sharing (AlBERT) and transformers without weight sharing (XLNet, DistilBERT etc.). We thank the reviewer for raising this point, and have made this clearer in the latest version of the paper.\
> __2. Re: The entire process of pruning is a bit vague and hard to replicate. Would it be possible to describe the whole process in pseudo code? (Is Algorithm 1 the whole process?)__ \
> We thank the reviewer for pointing out that our description of the pruning strategies was hard to follow. We have added detailed descriptions of the different strategies, including illustrations of the pruning strategies and algorithms, in Appendix A, in order to make them easily understandable without exceeding the page limit. We hope this makes the process clear and reproducible.\
> __3. Re: I think the authors should also describe the computational cost (or maybe wallclock time) required to perform the proposed pruning processes. It seems to me that the search space is rather large and requires a considerable amount of computation.__ \
> We have added details about the computational cost of fine-tuning to the Results section of the paper. In summary, it requires multiple fine-tuning passes. On tasks with large training data, we compute the thresholds based on a smaller subset of the target data; for these tasks, we randomly sample a small subset of the training data (<20% from each label) to fine-tune the model, and a validation set (~15% of the training set) to characterize significance. We find empirically that doing so results in the same elements getting pruned and approximated as when the entire training data is used. We further see that this subsampling is robust across models; if the reduced dataset works for one model, it works for all other models. Finally, each pass is expected to become progressively faster, since our iterative method potentially eliminates a component in each pass, and our ordering of elements ensures that time-consuming components are eliminated early. Therefore, the overheads at fine-tuning time are greatly reduced, allowing us to better explore the search space.\
> __4. Re: p.5 … we prune the element only if the training/validation loss. I think you should be more specific here. How did you actually use both the training and validation loss? Why do you need to look at the training loss when you are interested in the generalization error?__ \
> We thank the reviewer for pointing this out. We realize that this is not standard practice, and requires greater justification. We do not use both the training and validation loss for all tasks. However, the nature of datasets for the different GLUE tasks necessitates the use of both losses to produce stable models that meet the user’s constraints. When reporting results on the development set, for the larger datasets (like MNLI), we create a validation set using a small subset (about 15%) of the training data. We use the loss on this set to characterize the significance. However, on the smaller datasets (like WNLI), there isn't enough data to get a meaningful training-validation split. Hence, we need to directly use the loss on the training set to characterize the significance. When reporting results on the test set, we use loss on the development set to characterize significance. However, in this case, some datasets like QQP and WNLI have different distributions for development and test sets [Ref: https://gluebenchmark.com/faq]. Therefore, to ensure stability of results, we use both training and validation losses to characterize the significance of each component for these tasks. In particular, only if the training loss after removing the component is less than the (training) pruning threshold (defined as a function of the baseline training loss) *and* the validation loss after removing the component is less than the (validation) pruning threshold (defined as a function of the baseline validation loss), we remove the component from the pre-trained model. We will add this clarification to the latest version of the paper.\
> __5. Re: p.5 … weight groups of (Wn) … Why is this Wn? I thought this should be W.__ \
> We thank the reviewer for pointing out this mistake. We have corrected it in the latest version of the paper, and apologize for any confusion it caused.

---

### Official Review · AnonReviewer3 · 2020-10-29
**A new framework for pruning fine-tuned BERT by ensembling multiple tricks**

**Rating:** 5
**Confidence:** 4

**Review:**

After reading the rebuttal, some of my concerns are addressed by the additional experiments. But I also agree with other reviewers that the result is not very surprising. As R4 mentioned, the proposed method depends on the a specific downstream task where the "small" "general" BERT can be further pruned. For a fair comparison to previous work, baselines that are applied to a specific fine-tuning task need to be compared.

=====

This paper presents a new framework for creating small fine-tuned pre-trained language models.
The framework has 3 components:
1. a set of transformer components to be pruned
2. a significant analysis for identifying unimportant elements.
3. a set of techniques to prune or approximate the transformer element.

Pros:
1. The framework is very adaptive by considering different basic elements of the transformer.
2. The framework is very efficient by removing large components (e.g., layers, attention blocks, ffd layers) at first and small components (e.g., weight group) later.
3. The framework gathers multiple different pruning/approximation techniques and tries to explore the limit of pruning pre-trained models, which is appreciated.

Cons/Questions:
1. Is the loss used in significant analysis computed using the development set? If the validation loss is used, the experiment results in Table 1 are not reliable.
2. There are many BERT pruning papers. Providing comparison to these papers is very important to evaluate the proposed method. Can the model prune more weight at the same performance level? or  Can the model perform better at the same pruning ratio?
3. It is also helpful to present how much computing resource is needed to prune the network. E.g., how many prune-finetune cycles are needed.
4. Lack of results of pruning BERT-base on GLUE, which is a very standard and common setting.
5. In Figure 3, why Q8BERT + Speed Focus is even larger/slower than Q8BERT? With the same speed, Q8BERT + Speed Focus is significantly worse than Q8BERT.

Minor:
Page 5: less the minimum loss seen ==> less 'than' the minimum loss

---

> ### Author Response · Authors · 2020-11-20
> **Response to Reviewer3 (Part 1/2)**
>
> We thank the reviewer for their comments. We have addressed the individual concerns below. \
> __1. Re: Is the loss used in significant analysis computed using the development set? If the validation loss is used, the experiment results in Table 1 are not reliable:__ \
> The loss in significance analysis is **not** computed on the development set. We realize that this is a very important point that we failed to clearly address in the main paper. We have addressed this in the latest version. When reporting results on the development set, for the larger datasets (like MNLI), we create a validation set using a small subset (about 15 %) of the training data. We use the loss on this set to characterize the significance. On the smaller datasets (like WNLI), there isn't enough data to get a meaningful training-validation split. Hence, we directly use the loss on the training set. When reporting results on the test set, we use loss on the development set to characterize significance. We never report results on the same data that is used in significance analysis. We have clarified this in the Experiments section in the revised manuscript. \
> __2. Re: There are many BERT pruning papers. Providing comparison to these papers is very important to evaluate the proposed method. Can the model prune more weight at the same performance level? or Can the model perform better at the same pruning ratio?__ \
> We have added comparisons to previous works (Table 3) in the latest version of the paper. A majority of previous works for improving efficiency of Transformers directly pre-train efficient models from scratch. We evaluate the most commonly used techniques used to create efficient models, and demonstrate that our techniques are complementary, since these “efficient networks” still need to be fine-tuned for different downstream tasks, providing opportunities for optimization. In addition, we show that our techniques are also complementary to Q8BERT, a post-training quantization method, resulting in models that are up to 2.26x faster and up to 2.15x smaller than this already-optimized baseline. Compared to layer-dropping techniques that do not require training from scratch, we find that our framework produces models that are up to 3.65x faster or up to 8.82x smaller than Poor Man’s Bert, at iso-accuracy. \
> __3. Re: It is also helpful to present how much computing resource is needed to prune the network. E.g., how many prune-finetune cycles are needed.__ \
> We have addressed this in the latest version of the paper in the Results section. In summary, it requires multiple fine-tuning passes. On tasks with large training datasets, we compute the thresholds based on a smaller subset of the target data; for these tasks, we randomly sample a small subset of the training data (<20% from each label) to fine-tune the model, and a validation set (~15% of the training set) to characterize significance. We find empirically that doing so results in the same elements getting pruned and approximated as when the entire training data is used. We further see that this subsampling is robust across models; if the reduced dataset works for one model, it works for all other models. Finally, each pass is expected to become progressively faster, since our iterative method potentially eliminates a component in each pass, and our ordering of elements ensures that the more time-consuming components are eliminated early. Therefore, the overheads at fine-tuning time are greatly reduced, allowing us to better explore the search space.

---

> > ### Author Response · Authors · 2020-11-20
> > **Response to Reviewer3 (Part 2/2)**
> >
> > __4. Re: Lack of results of pruning BERT-base on GLUE, which is a very standard and common setting.__ \
> > Due to the page limit, we had provided results using BERT-base on GLUE in the appendix, and presented results on more efficient and/or more accurate networks (AlBERT, DistilBERT, Q8BERT etc.) in the main paper. However, we agree with the reviewer that this is a standard benchmark, and results need to be presented in the main paper itself. The latest version of the paper addresses this (Table 1), showing that our framework produces models that are 2.17x faster or 8.5x smaller than BERT-base using speed focus and size focus, respectively. \
> > __5. Re: In Figure 3, why Q8BERT + Speed Focus is even larger/slower than Q8BERT? With the same speed, Q8BERT + Speed Focus is significantly worse than Q8BERT.__ \
> > We thank the reviewer for pointing this out, and apologize for the confusion this caused. We had accidentally used the wrong label for the x-axis in Figure 3. Instead of saying Model Size/ Inference Time, it was wrongly listed as Model Size/ Speed, which made it seem as though Q8BERT + Speed Focus was slower than Q8BERT. In addition, the points on the graph were confusing, with two Q8BERT points - one indicating that it requires 100% inference time, and the other indicating that it has 25% the model size of BERT-Base. Q8BERT + Speed Focus is in fact significantly faster (1.61x) than Q8BERT at the same accuracy. We have improved the figure to be clearer in the revised manuscript.

---

### Official Review · AnonReviewer1 · 2020-10-30
**A simple yet impactful optimization approach using a suite of pruning methods.**

**Rating:** 6
**Confidence:** 4

**Review:**

The main goal of this paper is to introduce a simple methodology for optimizing transformer based models for efficiency and effectiveness.

The paper introduces two main ideas:

1)A top-down strategy for pruning components of a transformer model: Given a specific focus, say speed, the strategy is to consider pruning large coarse-grained components first followed by smaller finer-grained components. The pruning decision is made based on a “significance analysis” -- a component is considered significant for pruning if it from the model does not result in a substantial increase in the model’s loss (as decided by a pruning threshold).

2) Pruning and approximating techniques for different components: For example feed-forward networks are pruned by removing weights in groups (determined via a hyperparameter). For approximating self-attention a sign-matching technique for deciding which top K keys to use for computing Query x Key dot products.

The main strengths of this work are as follows:

1) The techniques do not require training networks from scratch and can be applied directly during fine-tuning.

2) The techniques are simple and should apply widely to most transformer-based models.

3) The empirical results support the claim that the technique can yield significant speed-up and memory-reductions while maintaining accuracy and even provide improvements in accuracy if that is the pruning goal. They show that technique is orthogonal to other models explicitly designed for speed and memory footprint (Q8BERT, DistillBERT) and can provide further improvements in both efficiency and effectiveness.

4) This is a practical and useful approach that should be widely applicable along with many useful insights about optimizing transformer-based systems.

I appreciate that the experimental results are reported with averages across multiple runs!

I don’t see any major weaknesses in the paper. Here are some areas that can be improved:

1) The description of the pruning strategies was hard to follow and needed to be tightened up. Possibly adding equations and some pseudo-code to the description should help.

2) I am curious to know what components get pruned cross the different models that were optimized. I wonder if there are systematic differences between original and distilled models and between auto-regressive (GPT) and auto-encoding style models.

3) Also some level of ablation analysis on the strategies used will be helpful. For example if the elements were not ordered based on the granularity would the results be any different? Since this is an iterative strategy the order should play an important role in selection and utility of the subsequent pruning steps. Same goes for the set of pruning strategies. A related question would be what gives the biggest gains.

4) What is the impact on the fine-tuning time? The baseline only requires one fine-tuning pass. Does this method require multiple fine-tuning passes? Or can the loss thresholds be computed on a smaller subset of the target data? This may be a good future work to look into for tasks where the training data is relatively large, where one cannot afford to exhaustively search through all the pruning strategies.

---

> ### Author Response · Authors · 2020-11-20
> **Response to Reviewer1**
>
> We thank the reviewer for their comments. We have addressed the individual concerns below. \
> __1. Re: The description of the pruning strategies was hard to follow and needed to be tightened up. Possibly adding equations and some pseudo-code to the description should help:__ \
> We thank the reviewer for pointing out that our description of the pruning strategies was hard to follow. We have added detailed descriptions of the different strategies, including flowcharts and pseudocode, in Appendix A, in order to make them easily understandable without exceeding the page limit. We hope this makes the process clear and reproducible.\
> __2. Re: I am curious to know what components get pruned cross the different models that were optimized. I wonder if there are systematic differences between original and distilled models and between auto-regressive (GPT) and auto-encoding style models.__ \
> We find that the differences are more pronounced across different tasks than across different style models. For example, for sentiment analysis, long-range dependency information is not very important. Hence, for all models fine-tuned for sentiment analysis, we observe that components in later layers (closer to the output) are more likely to be pruned. \
> Across models, we only observe subtle differences. For example, we find that XLNet (auto-regressive) is able to learn important task-specific information earlier than BERT (non auto-regressive). Hence, we are able to drop more components (in earlier layers) in XLNet than in BERT, leading to more efficient models for inference. In DistilBERT (a distilled model), we find that there is a clear demarcation in linguistic knowledge across layers due to the reduced capacity of the model. This is evidenced by the fact that components in the top four layers are never pruned across all Language Understanding tasks, while the boundaries are more soft in the original models. It would be interesting future work to use this framework to understand the inner workings of these models at finer granularities by identifying the significant components for different tasks and models. We have added this discussion to Appendix D in the latest version of the paper.\
> __3. Re: Also some level of ablation analysis on the strategies used will be helpful. For example if the elements were not ordered based on the granularity would the results be any different? Since this is an iterative strategy the order should play an important role in selection and utility of the subsequent pruning steps. Same goes for the set of pruning strategies. A related question would be what gives the biggest gains.__ \
> We have added an ablation study to Appendix C in the revised manuscript to address these questions. In summary, if the elements are not ordered based on granularity, the difference in the accuracy, size and latency of the final fine-tuned models is negligible. However, it leads to two orders of magnitude increase in the fine-tuning+optimization time, since we need to characterize the significance of a much larger number of parameters. In terms of pruning strategies, pruning the most time-consuming/ parameter-intensive blocks gives the biggest gains. We have added this analysis to Appendix C.\
> __4. Re: What is the impact on the fine-tuning time? The baseline only requires one fine-tuning pass. Does this method require multiple fine-tuning passes? Or can the loss thresholds be computed on a smaller subset of the target data? This may be a good future work to look into for tasks where the training data is relatively large, where one cannot afford to exhaustively search through all the pruning strategies.__ \
> We have added details about the fine-tuning time to the Results section of the paper. In summary, it requires multiple fine-tuning passes. As the reviewer mentions, this could be expensive for tasks with large training sets. To address this, we do in fact compute the thresholds based on a smaller subset of the target data; for these tasks, we randomly sample a small subset of the training data (<20% from each label) to fine-tune the model, and a validation set (~15% of the training set) to characterize significance. We find empirically that doing so results in the same elements getting pruned and approximated as when the entire training data is used. We further see that this subsampling is robust across models; if the reduced dataset works for one model, it works for all other models. Finally, each pass is expected to become progressively faster, since our iterative method potentially eliminates a component in each pass, and our ordering of elements ensures that more time-consuming components are eliminated early. Therefore, the overheads at fine-tuning time are greatly reduced, allowing us to better explore the search space.

---

### Decision · Program_Chairs · 2021-01-07
**Final Decision**

**Decision:**

Reject

**Comment:**

This paper introduces a set of techniques that can be used to obtain smaller models on downstream tasks, when fine-tuning large pre-trained models such as BERT. Some reviewers have noted the limited technical novelty of the paper, which can be seen more as a combination of existing methods. This should not be a reason for rejection alone, but unfortunately, the results in the experimental section are also a bit weak (eg. see [1-4]), there are not very insightful analysis and it is hard to compare to existing work. For these reasons, I believe that the paper should be rejected.


[1] DynaBERT: Dynamic BERT with Adaptive Width and Depth

[2] Training with quantization noise for extreme model compression

[3] MobileBERT: a Compact Task-Agnostic BERT for Resource-Limited Devices

[4] SqueezeBERT: What can computer vision teach NLP about efficient neural networks?